# SMS-Coastal, a New Python Tool to Manage MOHID-Based Coastal Operational Models

Fernando Mendonça [1,*] , Flávio Martins [1,2] and João Janeiro [1,3]

1 Centre for Marine and Environmental Research (CIMA), Campus de Gambelas, University of Algarve (UAlg), 8005-139 Faro, Portugal; fmartins@ualg.pt (F.M.); jmjaneiro@ualg.pt (J.J.)
2 Instituto Superior de Engenharia (ISE), Campus da Penha, University of Algarve (UAlg), 8005-139 Faro, Portugal
3 S2AQUA, Laboratório Colaborativo, Associação para uma Aquacultura Sustentável e Inteligente, 8700-194 Olhão, Portugal
* Correspondence: fmmendonca@ualg.pt

**Abstract:** This paper presents the Simulation Management System for Operational Coastal Hydrodynamic Models, or SMS-Coastal, and its novel methodology designed to automate forecast simulations of coastal models. Its working principle features a generic framework that can be easily configured for other applications, and it was implemented with the Python programming language. The system consists of three main components: the Forcing Processor, Simulation Manager, and Data Converter, which perform operations such as the management of forecast runs and the download and conversion of external forcing data. The SMS-Coastal was tested on two model realisations using the MOHID System: SOMA, a model of the Algarve coast in Portugal, and BASIC, a model of the Cartagena Bay in Colombia. The tool proved to be generic enough to handle the different aspects of the models, being able to manage both forecast cycles.

**Keywords:** operational oceanography; operational modelling; simulation management; ocean forecast; ocean modelling

## 1. Introduction

Operational Oceanography refers to a set of activities carried out to observe, describe, and predict the behaviour of the ocean in real-time [1,2]. It can be defined as a continuous process involving the collection, interpretation, and dissemination of measured data to analyse the ocean and forecast future conditions [3]. The field of operational oceanography consists of two main components [4]: observation systems, which assess physical and biogeochemical properties through in-situ monitoring and remote sensing [5,6], and modelling, which utilises ocean data to generate operational models for forecasting and producing data products [7].

Ocean modelling is a crucial activity within the scope of operational oceanography [8]. However, modelling coastal areas is particularly challenging due to their high variability, sudden changes, multiple influencing factors, socio-economic activities, data limitations, and scale considerations associated with coastal environments [9]. Addressing all these premises requires sophisticated modelling techniques, robust data collection strategies, and a comprehensive understanding of complex coastal dynamics. The Blue Economy, including ocean-related activities, plays a significant role in coastal areas. In Europe, coastal regions are densely populated and contribute more than 30% to the gross domestic product (GDP) [10]. Therefore, understanding coastal environments is essential, and operational coastal models can provide predictability, generating valuable data that can support sustainable economic growth while ensuring coastal protection [8].

Operating ocean forecasting models involves a series of routine actions. These actions include downloading forcing data from various suppliers, data processing, cropping, in-

terpolating, formatting, managing simulations, processing, archiving, and disseminating results, among others. Due to the repetitive nature of these tasks, automation is well suited for such operations. Automation systems with similar functionalities are already available for hydrological and watershed applications [11–13] as well as coastal modelling [14–16]. In this context, this article presents a novel software package called the Simulation Management System for Operational Coastal Hydrodynamic Models (SMS-Coastal). This system implements a generic framework designed to automate and manage all the processes involved in an oceanographic forecasting system.

This article is presented as follows: Section 2 introduces SMS-Coastal methods, designed to be as generic as possible, allowing their application to operationalise any given forecast model. Later, in Section 3, results are presented for two examples of operational modelling systems built using the MOHID Modelling System and managed by the proposed tool SMS-Coastal. Discussions and conclusions are presented in Sections 4 and 5, respectively.

## 2. SMS-Coastal Operation

The software package SMS-Coastal represents an innovative solution specifically built to effectively manage the various operations associated with daily simulations in operational oceanographic modelling systems. Developed to operate within the Windows operating system, SMS-Coastal is programmed using Python, a versatile, open-source, high-level, and object-oriented language. Python's extensive collection of adaptable modules and its clean code structure make it highly comprehensible and easily learnable [17]. These factors contribute to Python's popularity, as evidenced by its thriving and rapidly expanding community of users engaged in a wide range of applications, including traditional fields as well as cutting-edge domains such as artificial intelligence, data science, and machine learning [18].

Python has gained significant popularity within the scientific community due to its remarkable features, making it highly suitable for ocean modelling [19]. In addition to its extensive built-in functionalities, Python offers tools designed to handle common data formats used in ocean and atmospheric sciences, such as HDF5, GRIB, and NetCDF. It also provides support for OPeNDAP, tidal harmonic analysis software, and various scientific visualisation resources. The language possesses the necessary capabilities to execute essential tasks involved in maintaining continuous forecast cycles, ensuring the smooth operation of the modelling system. These tasks primarily involve controlling model inputs and outputs. Given these advantageous characteristics, the choice of Python as the programming language for SMS-Coastal is well justified.

As pointed out by Marta-Almeida et al. in [19], the execution of a single forecast cycle encompasses several essential steps: (1) Ensuring the availability and retrieval of required external forcing data, which typically include information regarding tides, freshwater discharges, external hydrodynamic fields, atmospheric surface heat and momentum fluxes, among other factors; (2) Effectively managing the archiving of data; (3) Conducting interpolation of external data onto the model grid; (4) Formatting the data to meet the specific requirements of the employed model; (5) Verifying the availability of initial conditions obtained from restart files generated in the preceding simulation cycle; (6) Generating model input files; (7) Performing the simulations; (8) Confirming the completion of the simulations; (9) Managing model outputs and preparing restart files for subsequent forecast cycles; (10) Reformatting, disseminating, and archiving the obtained results; (11) Providing comprehensive reports to the user's supervisor through appropriate messaging channels. Furthermore, daily simulations can quickly produce a large amount of data. To maintain free storage space on the servers, it is highly recommended to define a data archiving policy and create model output databases in different locations or on different devices. It is also advisable to incorporate routine operations to overwrite and/or remove unnecessary input and output data between forecast cycles.

Considering those fundamental steps, SMS-Coastal's most basic architecture was designed with three main components, as shown in Figure 1. The first is the Forcing

Processor, responsible for downloading all the oceanic and atmospheric external forcing data, conforming it to the model grid (e.g., by interpolation), and formatting it to suitable model file types. The next component is the Simulation Manager, which coordinates forecasts and spin-up simulations and manages the outputs in a local database. The last component is the Data Converter, which is a set of supporting sub-modules that reshape simulation outputs, convert them to a different format, and archive them in a different location. In SMS-Coastal, these three components are completely independent and may be accessed individually. In the next subsections, these components are described in detail.

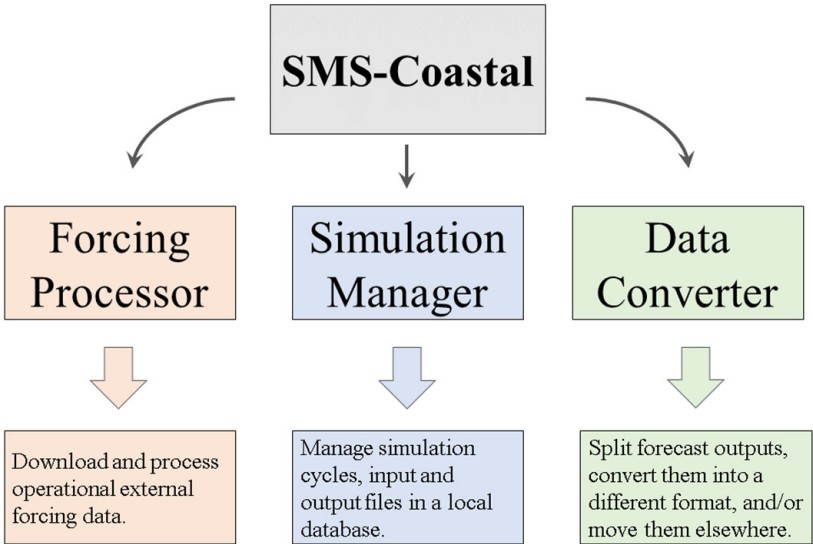

**Figure 1.** The three main components of SMS-Coastal are: The Forcing Processor (orange), the Simulation Manager (blue), and the Data Converter (green).

## 2.1. Forcing Processor

The Forcing Processor has the function of coordinating the operations required to prepare data from other global solutions for the model simulations. Due to the particularities of each data source, it was necessary to create a module in SMS-Coastal that would serve as a library with a set of specific methods appropriate to each data source. Nevertheless, the Forcing Processor main module performs the same generic sequence for all sources, as described by the structure chart in Figure 2. In addition to that, it runs the specific methods of that library, which currently supports the data to be used in the operational systems presented in Section 3, namely: the Operational Mercator global ocean analysis and forecast system from the European Union-Copernicus Marine Service [20]; the Skiron from the Atmospheric Modelling and Weather Forecasting Group (AM&WFG) of the Department of Physics of the National and Kapodistrian University of Athens (NKUA) [21–23]; and the North American Mesoscale (NAM) Forecast System of the NOAA's National Centres for Environmental Information (NCEP) [24,25].

As shown by the operations in Figure 2, the process for each data source starts by checking the directory structure on disc. If necessary, SMS-Coastal creates the set of folders and the log file that will be used in the subsequent operations. This structure is fixed and is composed of folders for the download, conversion, interpolation, and simulation data. Except for the last, the tool will overwrite files from a previous run to prevent data accumulation on disc. After this preparation, SMS-Coastal backs up the downloaded files in a different folder if the provider does not maintain a long-term repository.

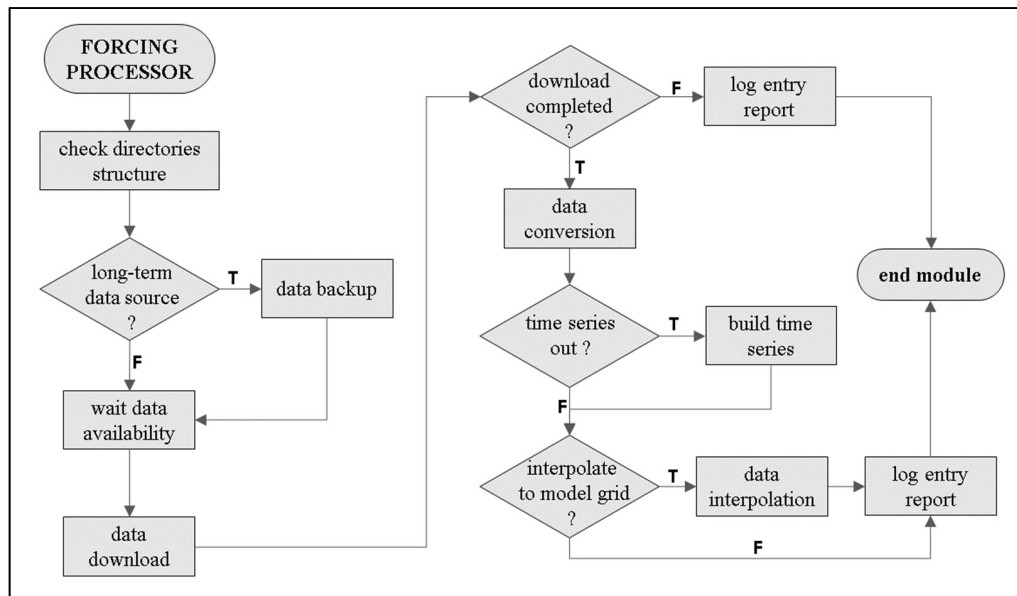

**Figure 2.** Generic scheme for SMS-Coastal Forcing Processor, in which the basic operations and sequence are the same for any data provider. The answers for a decision block in all figures with structure charts are given by the characters "T" for true and "F" for false.

Since each supplier has a specific schedule for uploading their data, SMS-Coastal configures the exact start time to trigger the download operation. In this way, SMS-Coastal will pause until the data is available. Additionally, when more than one source must be processed, the programme will queue the download operations, sorted by their start time. Depending on the data provider, the programme can retrieve the files directly from a web page, an FTP repository, or an OPeNDAP server. Other Data Access Protocols can be easily added to the system. If the download fails, the Forcing Processor is terminated for that specific source, and an error report is created. When two or more sources are scheduled for the same start time, the download order corresponds to the order of the inputs in the SMS-Coastal initialization file (see Section 2.3).

After the download, SMS-Coastal may convert the data to the format used by the hydrodynamic model being applied, such as from GRIB to NetCDF or from NetCDF to HDF5. If data is needed at just one point of the grid, the tool can be configured to convert it to a time series; otherwise, it can also manage the vertical and horizontal interpolation to the model grid. All files converted and interpolated within the Forcing Processor receive a well-defined name, so that when they are needed in a simulation operation, the Simulation Manager will easily find them. Before ending this component, the code keeps track of the operations for each data source in the created log file and can send an email report to inform on the success or error of the sequence.

### 2.2. Simulation Manager

SMS-Coastal is designed to handle two types of simulation cycles: one for spin-up simulations, the Restart Run, and another for operational forecasting, the Forecast Run. The Restart Run is used to initialise a model from a certain initial condition or from the data of larger-scale or global solutions. Once the Restart Run is completed, the model is in a more dynamically balanced state, which will serve as a better starting point for the actual forecast simulation.

The diagram in Figure 3 illustrates the simulation cycle process. The blue ribbons correspond to the Forecast Runs, and each one of them is a three-day forecast simulation. At the end of the simulation of the first day of each Forecast Run, the model writes the initial condition files that will be used the next day. At some point in the week, defined in SMS-Coastal's input file and on day i + 3 in the figure, as the initial conditions start to

degrade, a Restart Run (yellow ribbon) is executed in hindcast mode to adjust the model internal state to better dynamic conditions. Additionally, as shown in the diagram, both processes are executed simultaneously so that the prediction cycle is not interrupted. This is a representative example; times and durations can be easily adapted to other simulation requirements using the input method of SMS-Coastal (Section 2.3).

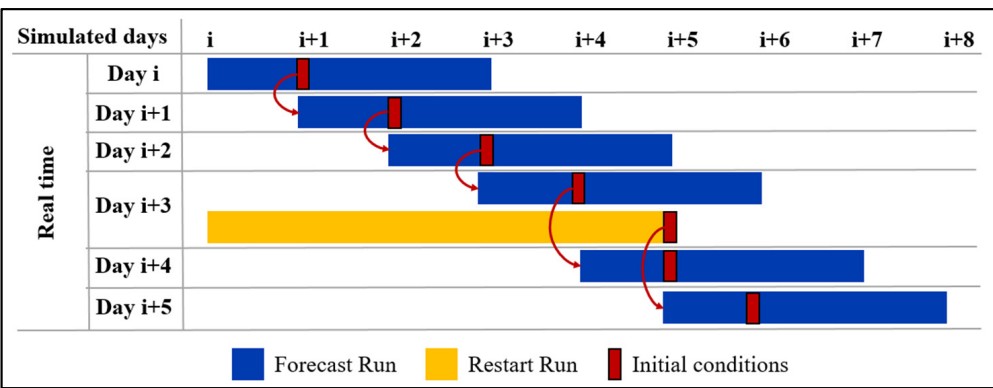

**Figure 3.** Forecast and Restart Run processes are being executed by SMS-Coastal. Every day the forecasting process is started, and after one day of simulation time, the initial conditions files for the next day's simulation are generated. Each prediction uses the past simulation files until a Restart Run is performed to produce fresh initial condition files.

### 2.2.1. Forecast Run

The operations sequence conducted in a Forecast Run is summarised in detail in the structure chart in Figure 4. Similar to the Forcing Processor, the programme checks the directory structure and prepares the simulation environment by removing all previous run outputs and gathering the essential files. The initial conditions are obtained from the files generated by previous simulation cycles, as explained in item 2.2. However, whenever a forecast and a spin-up cycle generate initial data for the same date, SMS-Coastal will give preference to those of the Restart Run. This aspect can also be seen in the diagram in Figure 3, at day i + 5, when the Forecast Run uses the files generated by the Restart Run of day i + 3.

When searching for forcing data, the tool selects the most recent one produced by the Forcing Processor component. In this way, SMS-Coastal keeps the forecast operational and on time, even if that component fails to obtain newer data. Furthermore, it makes an adjustment of the forecast range to the maximum possible by comparing the time outputs of the selected forcing data with the range defined in the programme inputs.

After the data setup, SMS-Coastal copies and writes the model configuration files, initiates the hydrodynamic model executable, and waits until its completion. If the forecast simulation fails, it writes the error record in the log file and exits the module. If it succeeds, all output files are copied to a local database. By default, the Data Converter component is launched to prepare files for archiving in an external database and for use in conversion operations. However, as the components are independent from each other, this feature can be disabled. At the end, the code writes a success message in the log and ends the module.

### 2.2.2. Restart Run

This step is described from the perspective of a model spin-up. An alternative to this could be the analysis cycle of a data assimilation scheme. Essentially, the result will be the same: to provide fresh initial conditions for the next forecast cycle. This methodology is very common in coastal and local models when there is a reliable global base solution in the background. The spin-up process follows a methodology very similar to that described in [26].

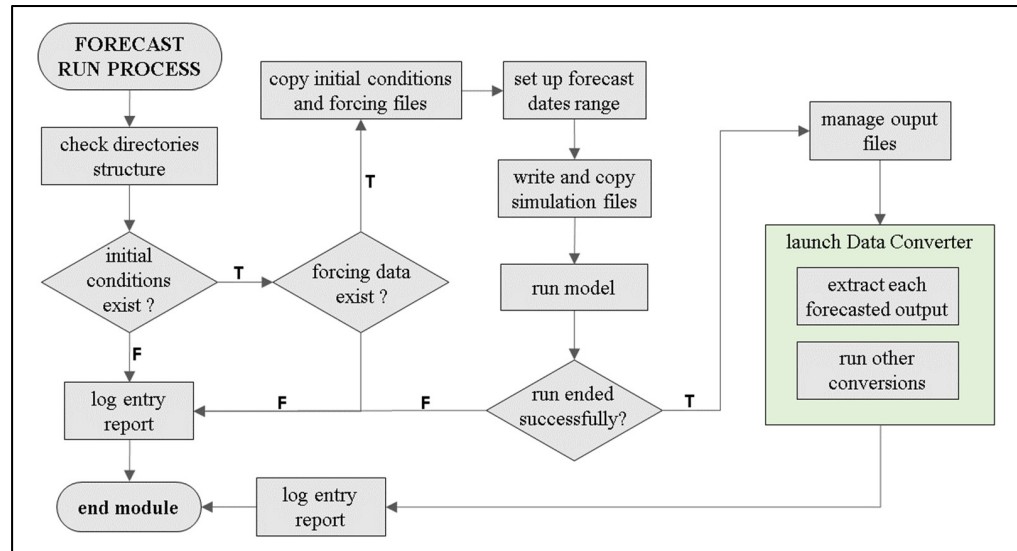

**Figure 4.** Forecast Run module's operation sequence. Checking the simulation environment comprises removing data from the previous simulation, creating the necessary folder structure, and preparing the log file. After this, the programme will check the existence of the initial conditions and external forcing data, terminating the module if any data is missing. With the forcing data, it will define the forecast date range to run the hydrodynamic model. If the simulation succeeds, the output files will be saved in a local database, and conversion operations will be launched inside the Data Converter component.

The sequence of operations performed in the Restart Runs shares some similarities with the Forecast Runs, as shown by the structure chart in Figure 5. There are, however, two major differences: the first is that there is no need to search for initial conditions files from a previous simulation since the model initiates from external initial and boundary conditions. The second is that the runs can be performed in stages with a gradual increase in time to prevent simulation instability. Due to this, SMS-Coastal does a repetition cycle to run each of the stages as shown in the figure, from stage 1 to "n" (n ∈ N, n > 0).

Considering the extensive duration of restart simulations, which encompass a hindcast spanning several days (refer to Figure 3), and that that is performed with gradual step time, it is important to acknowledge that the execution of all stages may exceed a single day in real time. In such cases, the generation of initial condition files for the last stage could experience a delay that renders them unusable for the Forecast Run of that particular day. To address this issue, SMS-Coastal incorporates an additional simulation stage to cover the period between the completion of the previous stage and the subsequent forecast, ensuring the provision of suitable initial conditions for the following Forecast Run. This aspect can also be seen in the diagram in Figure 3, where the Restart Run on day i + 3 did not generate initial conditions for the forecast of day i + 4, but for day i + 5.

### 2.2.3. Runs Manager

Considering that the forecast and spin-up processes are executed simultaneously, it was necessary to create the Runs Manager, the structure within the Simulation Manager that controls the Forecast and Restart Runs. It starts by running an instance of the Forcing Processor with the simulation dates as inputs to generate new forcing data. It then spawns a subprocess for the Forecast Run and another for the Restart Run. The latter, however, will only be executed on the day determined by the programme's inputs, e.g., once a week. The cycles must run in separate parallel subprocesses so that the prediction cycle will not be interrupted.

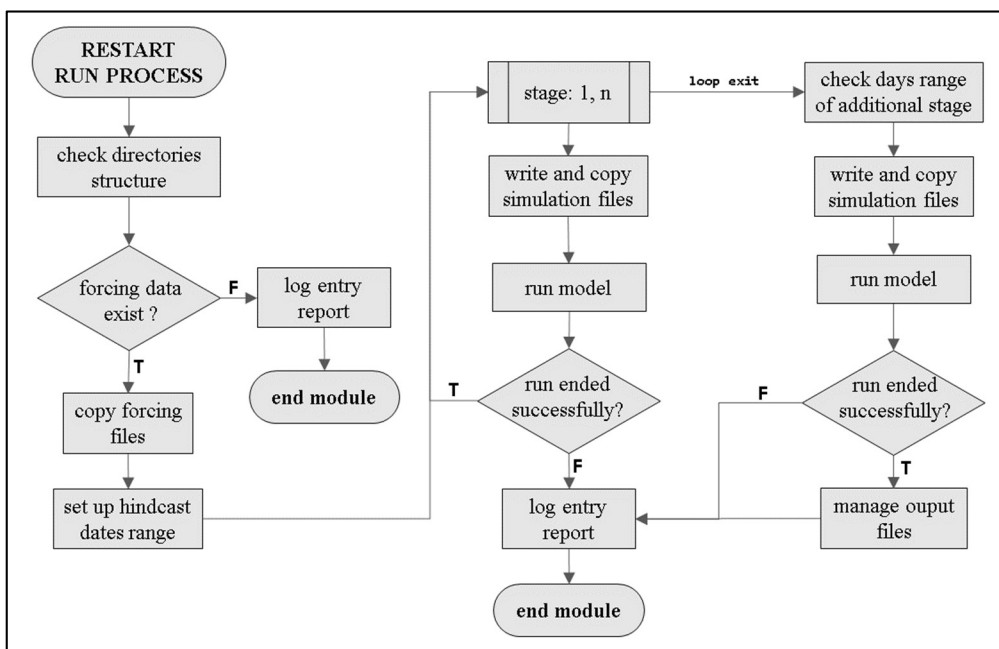

**Figure 5.** Restart Run module's operation sequence. The environment is set up in the same way as in a forecast, as is its time range. However, it does not need initial data from a previous simulation, and the restart can consist of more than one simulation or be conducted in stages that are conducted in a loop. After the final stage, an additional simulation is performed to generate initial condition files for the next Forecast Run.

### 2.3. Initialization File

SMS-Coastal was mainly designed to automate all the operations involved in producing coastal models' daily forecasts. For this reason, the Simulation Manager can also run instances of the Forcing Processor and Data Converter components. However, since they are independent of each other, the programme's initialization file can be configured to select specific functions and use SMS-Coastal for other purposes, e.g., process forcing data with the Simulation Manager disabled; run a simulation operation with the Forcing Processor disabled, making the Simulation Manager fetch previously processed data.

The initialization file is a text file that receives all user inputs to configure how and what SMS-Coastal should do. The programme's first operation is to read this file and check all inputs, which are a series of instructions organised in a similar system to Python dictionaries. In fact, the programme code converts these inputs into a dictionary, making the order of the entries irrelevant. Each file line has a keyword representing one SMS-Coastal execution variable and its value, which can be of any type (integer, float, or list) according to the key. Therefore, the execution variables supply information about which operation to run, the simulation date range, a list of oceanic and atmospheric external data sources, the number of model levels, when to perform the Restart Run, a list of e-mail addresses to send the reports to, what conversion operations to run, and other information. Examples of the input method for SMS-Coastal can be found on its GitHub page (see the Data Availability Statement section).

### 3. Results

The performance of SMS-Coastal is presented in this section using two different realisations of operational models using MOHID. Regardless, the tool can be upgraded to handle any other operational system. MOHID is a modelling system designed to model the marine environment with a modular architecture, programmed in ANSI FORTRAN, and features to simulate physical, chemical, and biological processes of the marine environment [27–30]. The two operational models used for testing SMS-Coastal are the Algarve Operational Modelling and Monitoring System (SOMA), a high-resolution operational

model for the Portugal south coast, and BASIC, the 3D operational model of the Cartagena Bay in Colombia. The associated operational systems have been operating for almost four years in SOMA's case and a little over three years in BASIC's.

SOMA is a 3D system that encompasses two increasing resolution levels: one with 2 km grid cells and another with 1 km grid cells. The details regarding the model's implementation can be found in [31]. It produces four-day forecasts of currents, salinity, and temperature and has been operational since 7 July 2019. Designed in a much smaller area, BASIC is a single resolution-level 3D model with 75 m grid cells, whose implementation details are in [32]. It generates three-day forecasts of currents, salinity, and temperature and has been operational since 1 May 2020. Each model runs in a virtual machine on the same server, sharing the same Intel(R) Xeon(R) Gold 6138 processor with a base frequency of 2.00 GHz. However, the resources allocated to each are not the same. SOMA has 10 virtual processors, 9.76 GB of RAM, and 450 GB of data space, while BASIC has 6 virtual processors, 8.00 GB of RAM, and 200 GB of data space.

Considering that SMS-Coastal was built to manage any MOHID-based operational system, it was necessary to develop a generic directory structure so that they could work together. In this case, the structure was mirrored for the two managed systems. The general structure is represented by the scheme in Figure 6. The folder and file names are fixed inside this structure for standardisation reasons. The contents of each simulation folder are as follows:

- FORC: folder used by the Forcing Processor to download, process, and store oceanic and atmospheric external forcing data.
- Hydrodynamic Model: contains the executable of the hydrodynamic model in use and all its necessary libraries and dependencies.
- Forecast: folder to be used by Forecast Run processes.
- Restart: folder to be used by Restart Run processes.
- SMS-Coastal.py: representation of all SMS-Coastal Python modules.
- init.dat: SMS-Coastal initialization file with all user inputs.

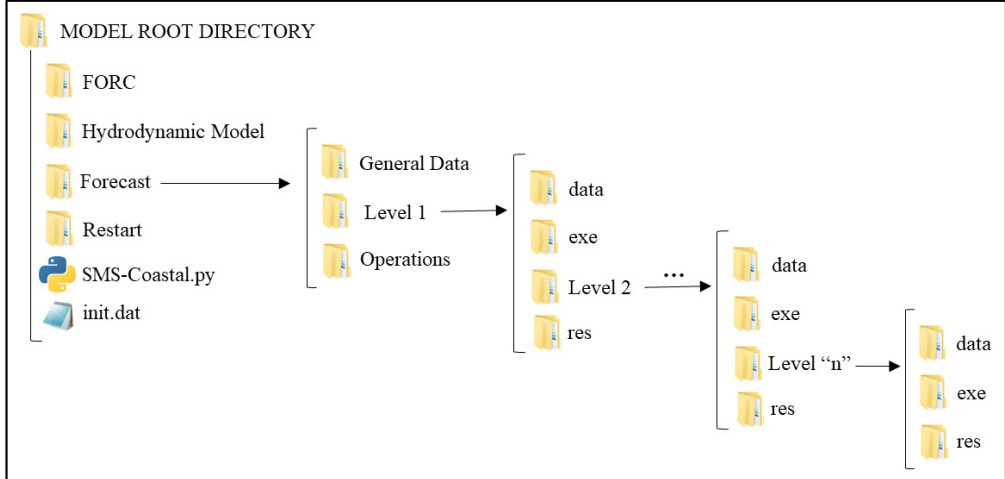

**Figure 6.** Schematic of the generic folder structure to be used by SMS-Coastal to manage an n-levels model. The root directory contains a folder for the Forcing Processor component operations, one for the hydrodynamic model files, and one for each simulation type, Restart Run and Forecast Run. The structure of the simulation folders is identical: "General Data" to store common input files for all levels; "Operations" as a local database for simulation output files; and one folder for each level of the model to store current simulation input and output files.

For each simulation type, there is a folder with the same internal structure, containing "General Data", "Operations", and the model's first-level folder. The first one is used by SMS-Coastal to store common input files between all levels, such as bathymetry data files, initial and boundary conditions, time series locations, tide information, and processed oceanic and atmospheric operational data. The programme will use the "Operations" folder to build the model's local database by placing in it all simulation outputs and log files, as well as formatted and converted results, as well as files in case of a failed run.

The folder structure demonstrated in Figure 6 fits an n-level nested project. Each level has three folders to store its respective input and output files: "data" for the set of data files with the user-defined parameters for the resolution of the hydrodynamic model governing equations; "exe" for the files related to the simulation execution, which is also the domain working directory; and "res" for all the domain's current simulation outputs. When applicable, it will have one more folder for the subsequent level. Part of this entire folder structure does not need to be previously defined, as SMS-Coastal, when executed, checks and creates all the missing structures, as is the case for the folders "FORC", "Operations", "exe", and "res" of each level.

Based on the log files of the SOMA and SMS-Coastal integrations, until the end of June 2023, over 1500 simulations were executed, of which 1518 were Forecast Runs and 234 were Restart Runs. On the other side, BASIC and SMS-Coastal integration accounted for 1351 runs, with 1211 Forecast Runs and 140 Restart Runs, until the same end date. Inevitably, not all runs were successful, considering that data providers, models, servers, and SMS-Coastal are susceptible to failure. Table 1 contains the failure statistics, that is, the simulations that did not finish successfully, for each model during this lifespan. Figure 7 depicts the graphical representation distributed in time of the stops presented in Table 1. In the table, the stops were classified into those caused by internal and external factors, including the following events:

- Code error: SMS-Coastal code optimisation is needed.
- Initial condition missing: SMS-Coastal was unable to find proper initial condition files from previous Forecasts or Restart Runs.
- Insufficient virtual memory: the simulation aborted due to a lack of space in computer random-access memory (RAM).
- Model error: simulation aborted by the hydrodynamic model.
- Server error: the virtual environment crashed.
- Forcing files missing: SMS-Coastal was unable to find suitable forcing files from providers.

**Table 1.** Failures observed for the models during the simulation management conducted by SMS-Coastal. The values for the Algarve coast model (SOMA) correspond to 1752 runs performed between July 2019 and June 2023, and for the Cartagena Bay model (BASIC), 1351 runs between May 2020 and June 2023.

| | | SOMA | | | BASIC | | |
|---|---|---|---|---|---|---|---|
| Stop Type | Stop Class | Stops | % of Stops | % of Runs | Stops | % of Stops | % of Runs |
| Code error | Internal | 25 | 19.4% | 1.4% | 6 | 7.5% | 0.4% |
| Missing initial conditions | Internal | 22 | 17.0% | 1.3% | 38 | 47.5% | 2.8% |
| Insufficient virtual memory | Internal | 9 | 7.0% | 0.5% | - | - | - |
| Model error | Internal | 30 | 23.3% | 1.7% | 2 | 2.5% | 0.1% |
| Server error | Internal | 7 | 5.4% | 0.4% | 5 | 6.2% | 0.4% |
| Forcing files missing | External | 36 | 27.9% | 2.1% | 29 | 36.3% | 2.1% |
| Total | | 129 | 100% | 7.4% | 80 | 100% | 5.8% |

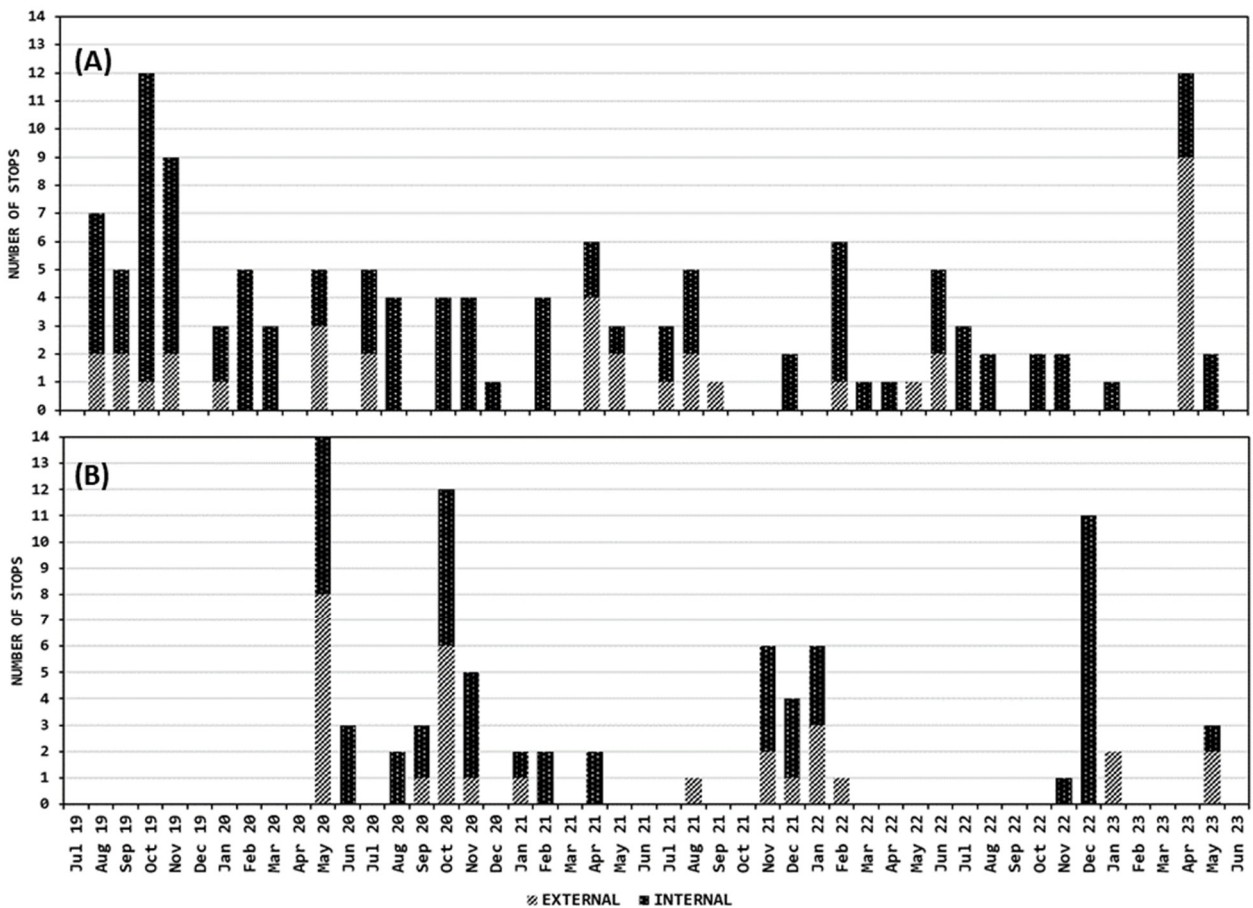

**Figure 7.** Failures over time for SOMA (**A**) and BASIC (**B**). As can be seen in B, since BASIC was launched only in May 2020, there are no stops computed for the time before that. No pattern of failure could be observed for the models. However, it was expected to have more stops at the beginning of each operational sequence since the SMS-Coastal generic code was being put to the test while managing the models.

In order to ensure the proper functioning of SMS-Coastal's generic code during simulation management, it is crucial to adhere to the prescribed structure of files and folders. Thus, to initiate a simulation operation, the model must be arranged as depicted in the root directory illustrated in Figure 6. The parameters of the equations should be stored within each respective "data" folder at their respective levels. Additionally, the necessary inputs must be appropriately specified in the initialization file named "init.dat". Once these steps are completed, the SMS-Coastal application can be executed. It is of utmost importance to accurately configure the input file, as the programme verifies all keywords and halts execution if any discrepancies are detected. However, in the context of executing continuous forecast cycles, the "init.dat" file only requires configuration once, ensuring subsequent operations can proceed smoothly.

## 4. Discussion

SMS-Coastal, a software designed to oversee operational cycles of coastal and oceanic models, was employed in this study to effectively manage forecast simulations for two MOHID-based systems. Over time, since the initial release of the programme, significant enhancements have been implemented in its source code, accompanied by the addition and removal of various modules.

One significant challenge of continuous simulation cycles is the limited availability of local storage space to accommodate the extensive volume of generated outputs. Within just a few days of execution, an enormous amount of data can accumulate. In the case of

SOMA, which encompasses two downscaling levels, a four-day forecast alone consumes approximately 8 GB of disc space. Given this aspect, it becomes crucial to devise an efficient data management strategy. As a result, several SMS-Coastal processes adopt the practise of overwriting outdated simulation files and integrating modules within the Data Converter to facilitate the transfer of the model's local database to an external storage system. This approach helps reduce storage constraints and ensures the continuation of operational simulations.

As exposed in Section 2.2.3, a simulation operation will split SMS-Coastal execution into more than one process. This is the case with a Restart Run, which must start in parallel with the Forecast Run. Therefore, as the runs are concomitantly handled by the multiprocessing Python built-in module, they are placed in separate folders inside the project directory (Figure 6). Furthermore, in our specific computer configuration, we observed a reduction in Forecast and Restart Run performances when both were running at the same time. Nevertheless, SMS-Coastal automatically monitors the end events of both processes to continue execution.

SMS-Coastal has maintained SOMA in operational mode since the beginning of July 2019. According to Table 1, the model was unable to complete 129 runs, which represents less than 8% of the total number of runs. Until November 2019, the larger number of internal failures, as shown by the (A) plot of Figure 7, resulted from problems in the programme's code that accounted in total for less than 20% of stops. This kind of interruption is always expected at the beginning of the implementation of SMS-Coastal into a new model. However, they can be viewed with optimism and used to improve the system by making it more robust and generic. The stops related to insufficient virtual memory are a specific cause of the model error stops, which were separately counted as they represent a situation that is related to SMS-Coastal. Newer versions of the management system were able to eradicate that problem.

BASIC has benefited from a more consolidated version of SMS-Coastal since, when its operational forecasting started, SMS-Coastal had already been used with SOMA for 10 months. For this reason, fewer code adjustments were necessary, and, as seen in Table 1, interruptions caused by programming problems represented less than 1% of total runs. The Cartagena Bay model had 1351 runs and was interrupted 80 times, or less than 6% of total runs. During this period, however, the model did not run for 10 days between October and November 2020 due to the suspension of the atmospheric data provider. BASIC has proved to be quite reliable since it has registered only two failures caused by numerical instabilities in the model. The significant increase in failures in December 2022 was due to the introduction of a new version of the code, which had some errors corrected as it was performing simulation cycles.

One major source of error in both models was the lack of external forcing data, caused mainly by the unavailability of the downloads. SMS-Coastal still has no other option but to abort a simulation if the required external data is missing, but future versions can be upgraded to retry the download after a waiting time. In Table 1, this class of stops was classified as an external factor; more than a quarter of all stops in SOMA runs were due to this factor, and in BASIC, the number is even higher, just over 36%. In the last model, several sequential stops of this type caused stops in the following days due to missing initial condition files. Together, they accounted for more than 80% of the total stops in BASIC. This fact is evident in the (B) plot of Figure 7, for the months of May and October 2020.

## 5. Conclusions

This research introduces SMS-Coastal, a self-contained software that implements a generic framework for managing forecast simulations in MOHID-based operational coastal models. Simulation management encompasses coordinating diverse operations and file handling. To enable users to specify operation parameters for each specific system, the software adopts a method of reading keywords from an initialization file. Consequently,

SMS-Coastal serves as a versatile tool for managing operational coastal and oceanic models, which is a critical step in supporting the Blue Economy.

The software was developed using the object-oriented programming language Python, which proved advantageous in simplifying the code. Reusable methods were employed to avoid repetitive coding and facilitate flexibility in different scenarios. Furthermore, the fundamental architecture of SMS-Coastal (refer to Figure 1) enables not only simulation operations but also result formatting and external forcing data processing, transforming it into a more versatile and multipurpose tool. However, it should be noted that SMS-Coastal currently does not support the operationalization of models with multiple nested models at the same level.

Despite the limitations it might present, the simulation management system presented in this work is prepared to make ocean and coastal models operational. SMS-Coastal, SOMA, and BASIC are now running in operational mode, producing daily forecast data. Nevertheless, the development of the programme remains an ongoing and iterative process. Regular updates and improvements will be consistently incorporated to enhance its robustness. Last, as part of future work, the integration of SMS-Coastal with an assimilation system could be assessed, serving as a viable alternative to the initial conditions generated by the Restart Run process.

**Author Contributions:** Conceptualization, F.M. (Fernando Mendonça), F.M. (Flávio Martins), and J.J.; methodology, F.M. (Fernando Mendonça); software, F.M. (Fernando Mendonça); supervision, F.M. (Flávio Martins) and J.J.; validation, F.M. (Fernando Mendonça); writing—original draft, F.M. (Fernando Mendonça); writing—review and editing, F.M. (Fernando Mendonça), F.M. (Flávio Martins), and J.J. All authors have read and agreed to the published version of the manuscript.

**Funding:** This work was supported by the Portuguese Foundation of Science and Technology (FCT) to CIMA [grant number UID/00350/2020 CIMA]; the EU-H2020 NAUTILOS project [grant number 101000825]; the COMPETE2020, NORTE 2020, and FCT, AEROS Constellation project [grant number AAC 04/SI/2019]; the ASTRIIS project [grant number 14/SI/2019-46092-ASTRIIS]; and the aid of a grant (#108747-001) from the International Development Research Centre, Ottawa, Canada, and the environmental authority CARDIQUE (special cooperation agreement for science and technology of 14 September 2018). Fernando Mendonça is a student, funded through the FCT Research Scholarships Programme [Reference UI/BD/153357/2022], to develop the PhD programme DATTCOM—Data Assimilation Tools Towards Coastal Operational Models.

**Institutional Review Board Statement:** Not applicable.

**Informed Consent Statement:** Not applicable.

**Data Availability Statement:** Data is contained within the article and SMS-Coastal source code is available on GitHub at the following link: https://github.com/fmmendonca/SMS-Coastal/, accessed on 1 August 2023.

**Acknowledgments:** The authors wish to give special thanks to CIMA-UAlg for providing the necessary conditions for the development of this work. We also would like to thank the following people for their support in this study: Antônio Augusto Neves, Eloah Rosas, Lara Mills, Marko Tosic, and Maria Mayrinck.

**Conflicts of Interest:** The authors declare no conflict of interest.

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
