# Peer review of "SMS-Coastal, a New Python Tool to Manage MOHID-Based Coastal Operational Models"

_jmse, doi:10.3390/jmse11081606_

Round 1

Reviewer 1 Report

English seems essentially ok

Reviewer 2 Report

This manuscript appears to be about the functionality and operation of a software, rather than a scientific paper. Nevertheless, the paper is well-written and clearly explains the author's intentions.

1 The software seems to be more like an automated plugin for an existing software, rather than a standalone program.

2 Is this software open-source and can the community benefit from it?

3 The core principles of this software are not introduced; it focuses more on explaining the operational aspects.

Author Response

The authors are confident that all the questions raised by the reviewer have been appropriately addressed, taking into account that the first reviewer also made similar observations. The new version of the manuscript incorporates several improvements in its text.

Point 1: "The software seems to be more like an automated plugin for an existing software, rather than a standalone program."

Response 1: While the authors recognise that SMS-Coastal cannot be applied to all hydrodynamic models, we see it as generic tool as its code is capable of managing two completely different systems, a coastal one (SOMA) and an estuary one (BASIC), both built based on the MOHID modelling system. We believe that the upgrades we've made in some points of the text made this aspect clearer in the manuscript.

Point 2: Is this software open-source and can the community benefit from it?

Response 2: We would like to thank the reviewer for this comment. Indeed, the source code for SMS-Coastal is publicly accessible on GitHub, with the link provided in the Software Availability section of the manuscript. While we acknowledge that the tool may have certain limitations, as stated in the conclusion: "Nevertheless, the development of the program remains an ongoing and iterative process. Regular updates and improvements will be consistently incorporated to enhance its robustness". The authors would like to add that, as it is open source, the tool can be edited and adapted to other coastal operational forecasting systems.

Point 3: The core principles of this software are not introduced; it focuses more on explaining the operational aspects.

Response 3: The article aims to introduce a new program and its working principle to manage operational systems with the premise that the modelling component of the system is already defined, calibrated and validated to the specific region. In the manuscript the management tool is described and then its performance is demonstrated using two existing coastal operational models.

The authors would like to thank the reviewer by the comments and suggestions and think they were important to produce a more clear and robust text.

Round 2

Reviewer 1 Report

The authors revised the manuscript and clarified many of my concerns. Some of the replies show the authors are not fully aware of some scientific issues regarding coastal modelling configurations. But maybe that is ok for this manuscript as the main focus is the description of an operational forecasting tool.

However, it is now clear to me that this paper is about a tool much more close to a "module" of MOHID than a generic methodology. For instance some tasks usually done by operational tools are here done by the model, so it is not straightforward to use the tool with other coastal model. Thus the title must be changed to reflect this. The title must clearly state the tool has been developed for MOHID, something like: SMS-Coastal, Methodology to Manage Coastal Operational Modelling with MOHID. This is important.

Author Response

Dear reviewer,

We would like to express our gratitude for the time and effort given in your feedbacks. As highlighted in our earlier revision, we do consider SMS-Coastal as a generic tool since it has the capacity to adapt to various implementations of the MOHID System. However, we acknowledge your valid concern about the possible ambiguity in the title. In line with your insightful suggestion, we have taken the decision to modify it to: "SMS-Coastal, a New Python Tool to Manage MOHID Based Coastal Operational Models", as it has been tested and is optimized to manage different systems based on the MOHID model, although in the future we intend to make SMS-Coastal a generic multi model platform.

Reviewer 2 Report

The manuscript can be accepted.

Author Response

Dear reviewer,
We would like to thank you for accepting our article for publication. However, we still made one last modification to the manuscript. We decided to change the title of the article to: "SMS-Coastal, a New Python Tool to Manage MOHID Based Coastal Operational Models", as suggested by the other reviewer, in order to resolve the ambiguity of the proposed tool's purpose.